

# Assessment and Monitoring of Land Degradation Using Geospatial Technology in Bathinda District, Punjab, India

Naseer Ahmad, Puneeta Pandey*

Centre for Environmental Sciences and Technology, Central University of Punjab, Bathinda, Punjab-151001, India

*Correspondence to*: Puneeta Pandey (puneetapandey@gmail.com)

**Abstract.** Land degradation leads to alteration in ecological and economic functions due to decrease in productivity and quality of the land. The aim of the present study was to assess land degradation with the help of geospatial technology - Remote Sensing (RS) and Geographical Information System (GIS)) in Bathinda district, Punjab. The severity of land degradation was estimated by analysing the physico-chemical parameters in the laboratory and correlating them with satellite

based studies. The results revealed that the soils in the study area were exposed to the salt intrusion which could be mainly attributed to irrigation practices in the state of Punjab. Most of the soil samples of the study area were largely found slightly or moderately saline with a few salt-free sites. Further, majority of the soil samples were calcareous and a few samples were alkaline or sodic in nature. A comparative analysis of temporal satellite datasets of Landsat-7 ETM+ and Landsat-8 OLI_TIRS of the year 2000 and 2014 respectively, revealed that the water body showed a slight decreasing trend from 2.46

$km^2$ in 2000 to 1.87 $km^2$ in 2014; while, the human settlements and other built-up areas expanded from 586.25 $km^2$ to 891.09 $km^2$ in a span of 14 years. The results also showed a decrease in area under barren land from 68.9847 $km^2$ in 2000 to 15.2602 $km^2$ in the year 2014. Significant correlation was observed between the Digital Number (DN) of near Infrared band and pH and EC. Therefore, it is suggested that the present study can be applied to projects with special relevance to soil scientists, environmental scientists and planning agencies that can use the present study as a baseline data to combat land degradation

and conserve land resources in an efficient manner.

**Keywords:** Remote Sensing (RS), Digital Number (DN), Geographical Information System (GIS), Calcareous, Alkaline

## 1 Introduction

The land use and land cover change (LULCC) effects have turn out to be a prime issue for the scientists concerned with global environmental change (Munoz-Rojas et al., 2015; Ochoa et al.,2016). The land use activities have a considerable

influence on the people posing serious consequences on social, economic, and ecological aspects of humans surrounding (Burchinal, 1989; FAO, 1997).

Land degradation is the process that makes land unsuitable for human beings as well as for soil ecosystems (Kimpe and Warkentin, 1998), occurs in arid, semi-arid and sub-humid areas as a result of anthropogenic activities and climatic variations (Barbero-Sierra et al., 2015) and eventually puts livelihoods and sustainable development at risk (Fleskens and



Stringer, 2014). It is the alteration in ecological and economic functions due to the decrease in the productivity and quality of the land (Hill et al., 2005) that leads to decline in the biological productivity of land due to climate change and human activities (Zhang et al., 2014). Land degradation poses a great threat to the food security and damages the environmental safety of land as well as influenced the sustainable development of society and economy (Zhao et al., 2013). Degradation of

soil is a desertification processes that leads to exhaustion of other natural resources in both developed and developing countries and affect arid, dry and even sub-humid areas (Omuto et al., 2014; Stringer and Harris, 2014). The degradation of soil happens not only as a result of interaction between physico-chemical and biological factors comprising topography, soil properties and climatic features (Brevik et al., 2015; Taguas et al., 2015), but also includes human factors, land use management practices (Khaledian et al., 2013; Camprubi et al., 2015; Costa et al., 2015; Cerda et al., 2016). Improper land

use practice has been attributed as one of the major causes of land degradation by various researchers (Biro et al., 2013; De Souza et al., 2013; Pallavicini et al., 2014; Mohawesh et al., 2015).The resilience and stability of landscape are affected to a great extent by the soil system which in turn affected by the inherent balance between inputs and nutrient loss and carbon (Amundson et al., 2015).

Land degradation is a severe problem due to which 1.5 billion people are threatened (Nachtergaele et al., 2010) and about

1·9 billion hectares of land and 250 million people are affected worldwide (Low, 2013). There is an increasing trend in severity of degradation, covering most of the world's land area which includes 30% forests, 20% cultivated areas, and 10% grasslands undergoing degradation (Bai et al., 2008). According to Barrett and Hollington (2006), approximately, 10 to 20 million populations live on the land affected by salts with poor productivity and under alarming threats of ecosystem destruction. Every year approximately 6 million hectares of agricultural land turn into unproductive due to various processes

of the soil degradation (Asio et al., 2009). A target of zero net land degradation (ZNLD) at RIO + 20 was set by United Nations Convention to Combat Desertification (UNCCD) held in Brazil in 2012, aimed at reducing the rate of land degradation and promoting the rate of restoration of already degraded land (Easdale, 2016).

According to an estimation of Indian Council of Agricultural Research (ICAR, 2010), about 120.40 million hectares (out of 328.73 M ha) of land in the country is affected by land degradation. In the state of Punjab, 2.33% (1172.84 km$^2$) of the land

area is considered as wasteland; the highest area under wasteland being of Muktsar district (186.8 km$^2$), followed by Ferozpur (148.1 km$^2$), Bathinda (144.4 km$^2$) and Gurdaspur (94.5 km$^2$).Change detection studies of LULC in an area have proven to be very effective in assessing the potential adverse impacts on environment (Leh et al., 2013). Hence, it becomes essential to devise effective strategies for land management at landscape level by analysing the extent of land degradation using model simulation studies for LULC dynamics (Gessesse et al., 2015).

Though the land area of Punjab under different wasteland categories is less as compared to other states (State of Environment Punjab, 2007) but the land degradation assessment and monitoring is very essential to improve understanding and assistance in decision-making processes. Hence, this research uses both remote sensing methods as well as physico-



chemical analysis of soil for assessing the severity of land degradation. The results obtained by both the approaches have been correlated to get a better picture of the extent of land degradation in Bathinda, a semi-arid town in western India.

## 2 Materials and Method

### 2.1 Study area

5  Bathinda is one of the historical and important cotton‐producing towns of North-western India, situated in the Malwa region of Southern Punjab. The total geographical area (TGA) of the district is about 336,725 hectares (3367.25 km$^2$) and lies between 29˚.33ˈ-30˚.36ˈ North latitude and 74˚.38ˈ-75˚.46ˈ East longitude. The study area shows typical condition of desertification and soil salinization; hence, effective means of combating soil salinization and desertification need to be pursued. From the study area, a total 21 sites (Figure 1) were selected for soil sample collection for the analysis of physico-

10  chemical parameters.

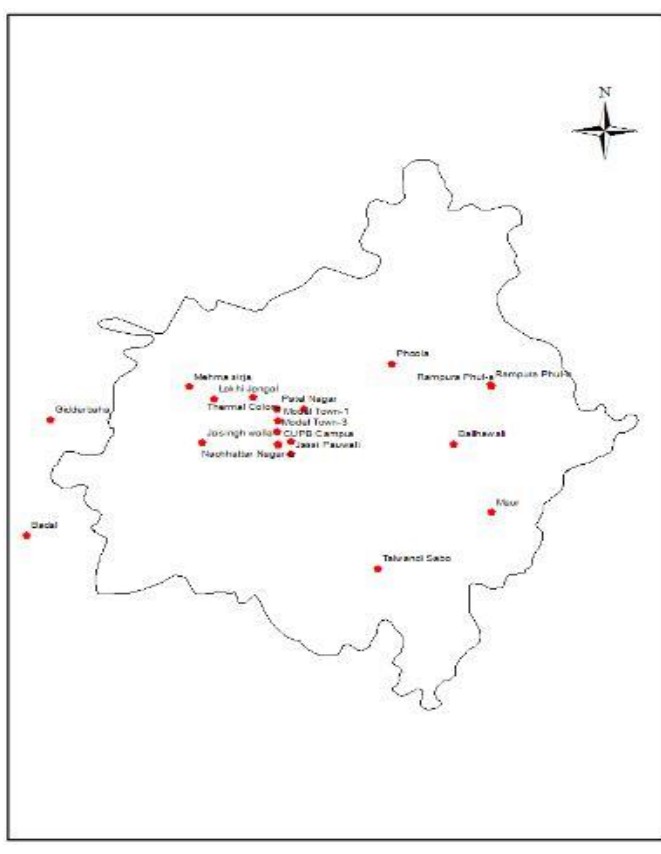

Figure 1: Sampling sites of the study area



## 2.2 Satellite data

The least-clouded multispectral Landsat satellite images of the year 2000 and 2014 (Table 1) were procured from United States Geological Survey (USGS) (*www.usgs.gov*) and Global Land Cover Facility (GLCF) (*www.glcf.umd.edu*).

Table 1: Details of the Satellite Images used for study

| Spacecraft ID | WRS: P/R | Acquiring Date | Dataset/ Sensor ID | Producer | Attri. | Type | Location |
|---|---|---|---|---|---|---|---|
| Landsat_7 | 2: 148/039 | 2000-12-25 | ETM+ | Earthsat | Ortho, Nv | GeoTIFF | India |
| Landsat_8 | 2: 148/039 | 2014-10-21 | OLI_TIRS | USGS | GLS-2000 | GeoTIFF | India |

## 2.3 Software used

Satellite image processing software, Earth Resources and Data Analysis System (ERDAS) Imagine (Hexagon Geospatial - formerly ERDAS, Inc.) and ArcGIS 10.1 (Environmental Systems Research Institute-ESRI product) were used to process, classify, analyse and display the satellite images.

10  ## 2.4 Land use land cover (LULC) map of Bathinda

The false colour composite (FCC) of multi-temporal Landsat satellite images of the year 2000 and 2014was generated on 1:50,000 scale. Further, unsupervised classification using ISODATA algorithm was carried out to gain an idea about spectral variability of classes. In this method, the separation of clusters of pixels was carried out based on statistically similar spectral response patterns, to gain information categories by determining classes that were spectrally different, and then defined their
15  information value (Lillesand and Kiefer, 1994). Finally, land use land cover map was generated by supervised classification using Maximum Likelihood Classifier (MLC) algorithm, wherein the pixels of unknown class were allocated to a particular land use class in which it had the highest likelihood of membership.

## 2.5 Physico-chemical Analysis of soil

For the present study, three physic-chemical parameters- pH, EC and Alkalinity were analysed for 21 soil samples in the
20  study area as represented in Table 2.



Table 2: Parameters of soil analysis

| S. No. | Parameter | Method | Instrument | Reference |
|---|---|---|---|---|
| 1 | pH | IS: 2720, part 1-1983 | pH meter (OAKTAN-PC 2700) | IS 1987 |
| 2 | Electrical Conductivity (EC) | IS: 14767-2000 | EC- Meter (Systronic water analyser 371). | IS 2000 |
| 3 | Alkalinity | IS 2035 (part 23)- Reaffirmed 2003 | Titration assemblage | Gupta, 2007 |

**2.5.1 pH:** The pH was determined in accordance with the procedure of IS: 2720, part 1-1983where pH meter (OAKTAN-PC 2700) was used to record the pH in an extract of 1:2 of soil or supernatant liquid.

**2.5.2 Electrical Conductivity (EC):** The measurement of Electrical Conductivity was done in accordance with IS: 14767-2000using EC- Meter (Systronic water analyser 371. An extract of soil sample or supernatant liquid of 1:2 soil water suspension was prepared and filtered using Whatman's filter paper to avoid any interference before recording the conductivity by EC-meter.

**2.5.3 Alkalinity:** With the help of Titration method, the Alkalinity of the soil samples was measured using 0.05N $H_2SO_4$, 0.5% Methyl red indicator and 0.25% phenolphthalein indicator. Since, the carbonates were absent in the solution, it did not turn into pink. It was then titrated till the color changed from yellow to rose red, indicated as the end point of titration. The concentration of bicarbonates was calculated from the formula as given in Eq. (1)

15 $$HCO_3^- g/L = \frac{\text{Normality of } H_2SO_4 \times \text{Vol.of } H_2SO_4 \times \text{Eq.wt.of } HCO_3^-}{\text{mL of aliquot taken}} \qquad (1)$$

**2.6 Correlation of Satellite data with physico-chemical parameters**

The Digital Numbers (DN) of the satellite image at the respective sampling sites was correlated with the pH and EC readings to gain an idea about the efficacy of satellite data in relation to laboratory analysis of soil samples.



# 3. Results and Discussion

## 3.1 LULC map of Bathinda

The False Colour Composite (FCC) image was prepared for the year 2000 Landsat ETM+ satellite data using "432 RGB" band combination (Figure 2a) while the band combination of "543 RGB" was used for the Landsat 8 satellite image (Figure 5 2b) for the year 2014.

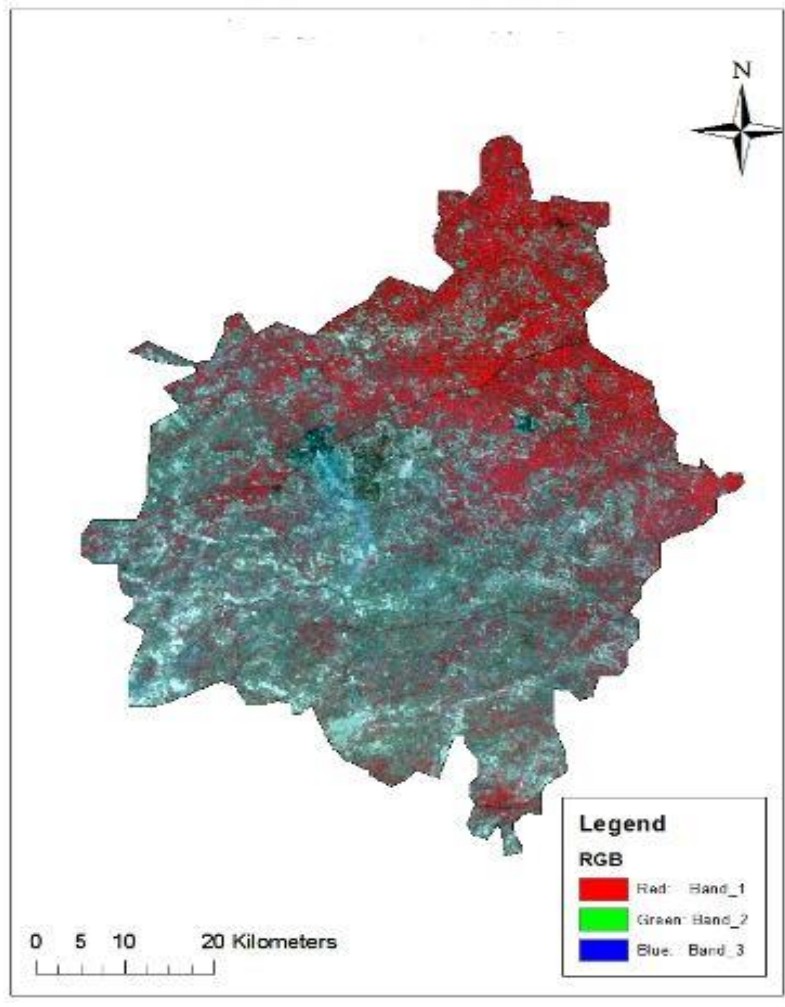

Figure 2(a): FCC of Landsat ETM+ of 2000




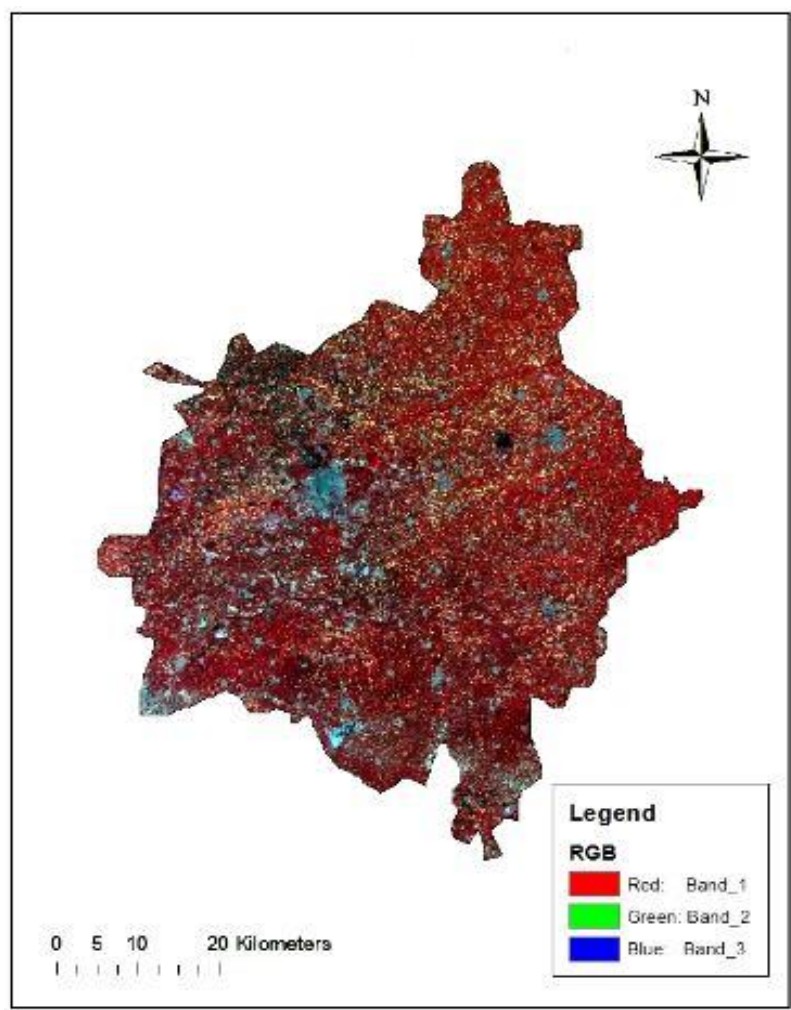

Figure 2(b): FCC of Landsat OLI-TIRS data of 2014

Followed by preparation of FCCs and visual interpretation, land use land cover maps of Bathinda were prepared for the year 2000 and 2014 as given in Figure 3(a) and 3(b) using Iterative Self-Organizing Data Analysis Technique (ISODATA).





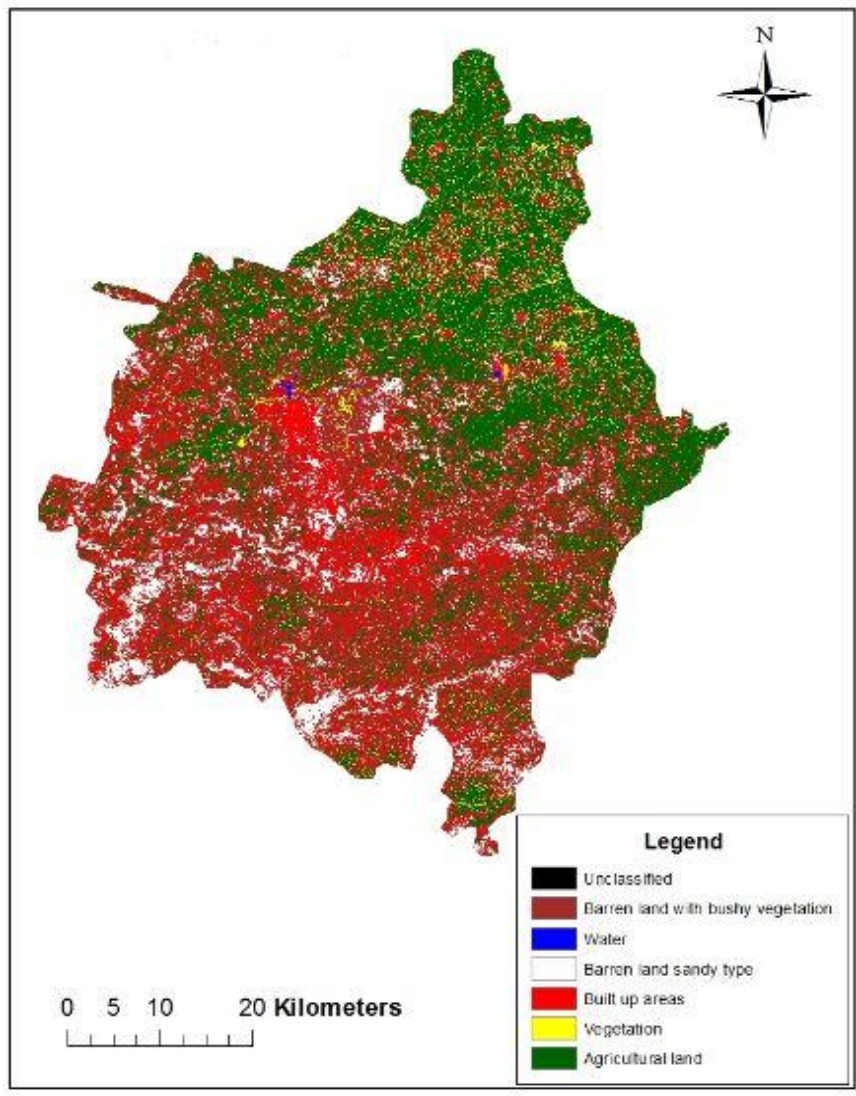

Figure 3(a): LULC map of Bathinda using unsupervised classification for 2000



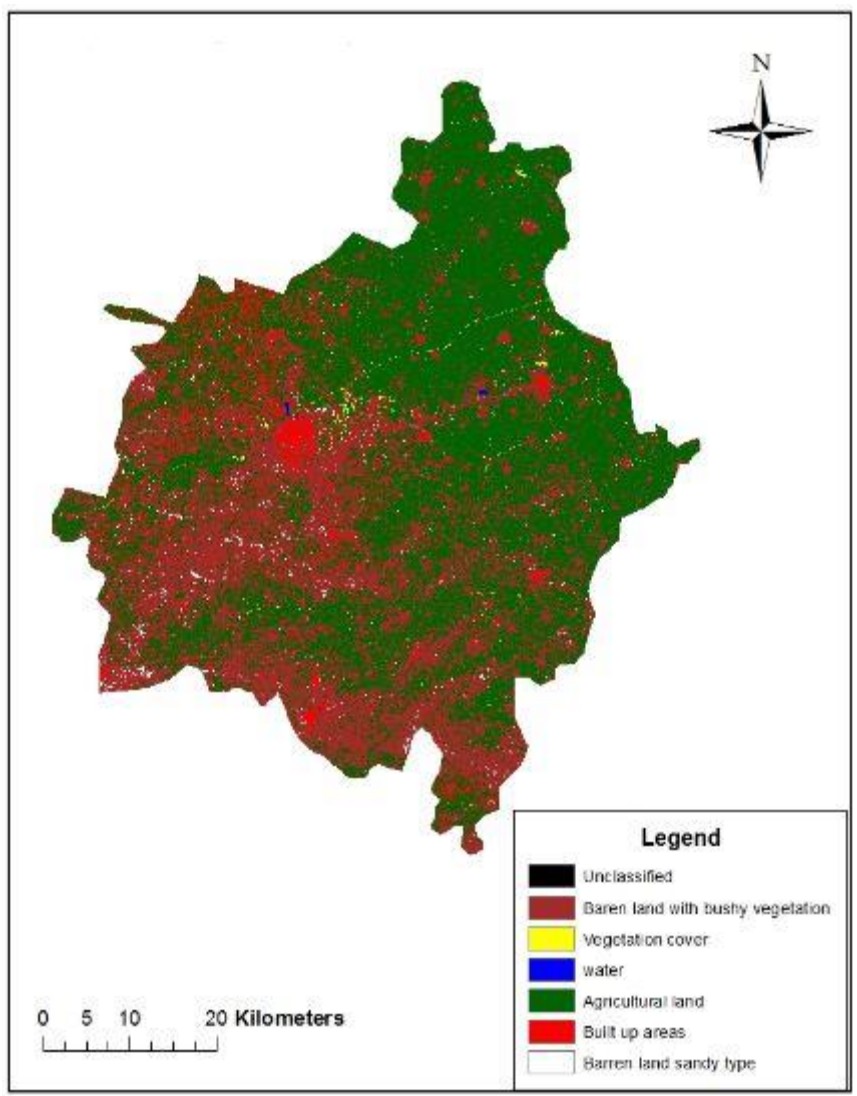

Figure 3(b): LULC map of Bathinda using unsupervised classification for 2014

After unsupervised classification, where an insight was gained about spectral variability of classes; supervised classification of False Colour Composite (FCC) images of both the temporal data sets of the Landsat image was carried out for the year 2000 and 2014 as given in Figure 4(a) and 4(b) using Maximum Likelihood Classifier (MLC) algorithm. The agricultural lands with crops and without crops were assigned with dark green and light green, respectively. The blue colour was assigned to the water bodies while the red and yellow colour was given to settlements and trees/forest cover, respectively.




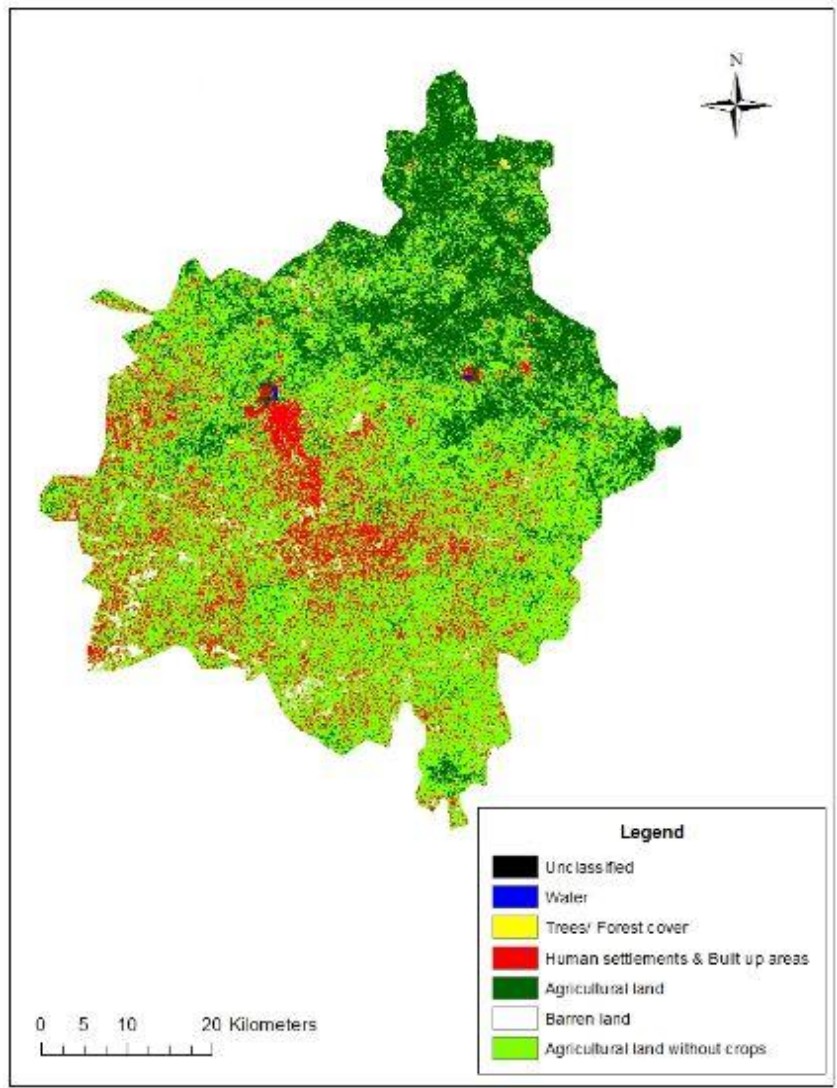

Figure 4(a): LULC map (Supervised classification) of Bathinda for 2000



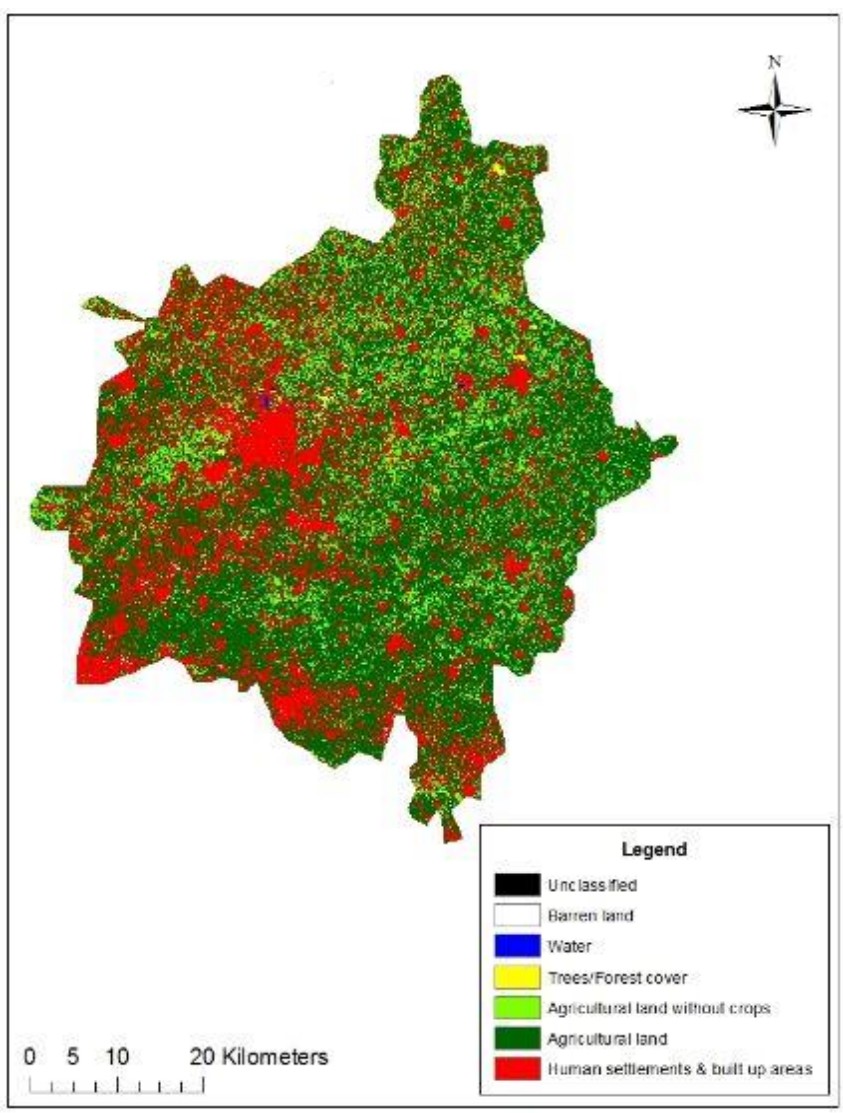

Figure 4(b): LULC map (Supervised classification) of Bathinda for 2014

## 3.2 Accuracy Assessment

The overall classification accuracy of the Landsat image (supervised classification) of the year 2000 was found to be 96.48%
5 and the overall Kappa Statistics was found to be 0.947 (Table 3(a)). Similarly, for Landsat image (supervised classification) of the year 2014, it was found to be 97.66% with Kappa Statistics of 0.964 (Table 3(b)). It indicated that the results were appreciably better than random of the values contained in an error matrix (Jensen, 1996).



Table 3: Accuracy Assessment Report

3(a) Accuracy Totals (year 2000):

| S. No. | Class Name | Reference Totals | Classified Totals | Number Correct | Producer's Accuracy | User's Accuracy |
|---|---|---|---|---|---|---|
| 1 | Unclassified | 115 | 114 | 114 | ----- | ----- |
| 2 | Water | 0 | 0 | 0 | ----- | ----- |
| 3 | Human settlements and built up areas | 10 | 13 | 9 | 90.00% | 69.23% |
| 4 | Agricultural land | 43 | 40 | 40 | 93.02% | 100.00% |
| 5 | Agricultural land without crops | 81 | 85 | 81 | 100.00% | 95.29% |
| 6 | Trees/Forest cover | 0 | 0 | 0 | ----- | ----- |
| 7 | Barren land | 7 | 4 | 3 | 42.86% | 75.00% |
| 8 | **TOTALS** | **256** | **256** | **247** | | |

Overall Classification Accuracy = **96.48%**

Kappa (K^) Statistics: Conditional Kappa for each Category was also obtained. The overall Kappa statistics was **0.947.**

3(b) Accuracy Totals (year 2014):

| S. No. | Class Name | Reference Totals | Classified Totals | Number Correct | Producer's Accuracy | User's Accuracy |
|---|---|---|---|---|---|---|
| 1 | Unclassified | 128 | 128 | 128 | ----- | ----- |
| 2 | Water | 0 | 0 | 0 | ----- | ----- |
| 3 | Human settlements and built up areas | 38 | 39 | 18 | 100.00% | 97.44% |
| 4 | Agricultural land | 67 | 70 | 66 | 98.51% | 94.29% |
| 5 | Agricultural land without crops | 21 | 18 | 17 | 80.95% | 94.44% |
| 6 | Trees/Forest cover | 1 | 1 | 1 | 100.00% | 100.00% |
| 7 | Barren land | 1 | 0 | 0 | 42.86% | 75.00% |
| 8 | **TOTALS** | **256** | **256** | **250** | | |

Overall Classification Accuracy = **97.66%**

Kappa (K^) Statistics: From the results the overall Kappa statistics (conditional Kappa for each Category of the year 2014) was **0.964.**



### 3.3 Change Detection Analysis

To analyse the change occurring between different land features for a period of 14 years, the supervised images of both the year 2000 and 2014 were used as input images and the change was highlighted as 20% increase (blue colour) and 20% decrease in the final map (pink colour) depicting change detection (Figure 5). Rest of the classes included unclassified, unchanged, some increased and some decreased which were pointed out in black colour. The increased portion predominantly indicated the expansion of settlements with little increase in vegetation. The decreased portion depicted the decrease in overall land area under agriculture (with or without crops).

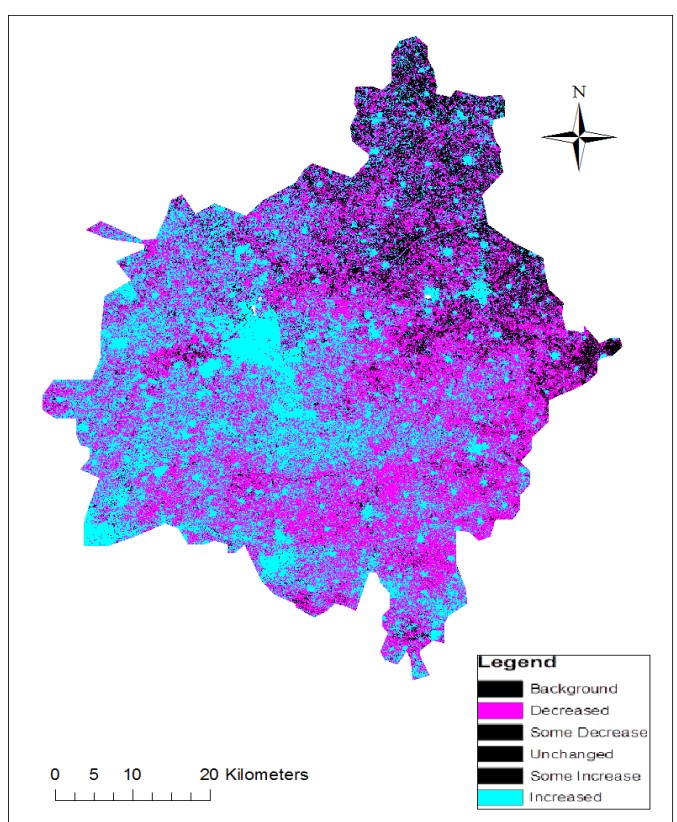

Figure 5: Change Detection Map of Bathinda district

Table 4 describes the change detection in total area (km$^2$) covered by different LULC between the year 2000 and 2014 procured from Landsat satellite data.

Table 4: Change detection in different LULC (km$^2$) between 2000 and 2014

| S. No. | LULC Class | Area (km$^2$) in 2000 | Area (km$^2$) in 2014 | Change Detection (km$^2$) between 2000-2014 |
|---|---|---|---|---|
| 1. | Water | 2.46 | 1.8783 | 0.58 (decrease) |



| 2. | Human settlements and built up areas | 586.25 | 891. 10 | 304.85 (increase ) |
|---|---|---|---|---|
| 3. | Agricultural land | 3002.23 | 2399.80 | 602.44 (decrease) |
| 4. | Trees/Forest cover | 29.43 | 45.73 | 16.30 (increase) |
| 5. | Barren land | 68.98 | 15.26 | 53.7243 (decrease) |

With reference to Table 4, the area under water body, human settlements and built-up areas, agricultural land, trees/ forest cover and barren land was found to be 1.87 km$^2$, 891.09 km$^2$, 2399.79 km$^2$, 45.73 km$^2$ and 10.35 km$^2$, respectively. The water body showed a slight decreasing trend from 2.46 km$^2$ in 2000 to 1.87 km$^2$ in 2014. The human settlements and other built-up areas expanded from 586.25 km$^2$ to 891.09 km$^2$ in the 14 years span of time.  This could also be attributed to increasing urbanization in the city of Bathinda during the last 14 years.

### 3.4 Laboratory Analysis for Physico-Chemical Parameters:

The values of pH, EC and Alkalinity of the soil samples collected from the study area are given in table no. 5.

Table 5: Results of physico-chemical parameters of soil

| S.NO. | SITE | pH | EC (dS/m) | Methyl Orange Alkalinity (HCO$_3^-$)  g/L as CaCO$_3$ | SOIL TYPE |
|---|---|---|---|---|---|
| 1 | JassiPauwali | 8.11 | 4.04 | 0.161 | Calcareous, Slightly saline |
| 2 | Talwandi Sabo | 7.78 | 2.63 | 0.161 | Calcareous, Salt free |
| 3 | Maur | 8.44 | 2.50 | 0.140 | Alkali or Sodic soils, Salt free |
| 4 | Rampuralphul- a | 8.20 | 8.78 | 0.161 | Calcareous or saline, Moderately saline |
| 5 | Rampuraphul- b | 7.84 | 8.17 | 0.122 | Calcareous, Moderately saline |
| 6 | Ballhawali | 7.90 | 5.89 | 0.109 | Calcareous, Slightly saline |
| 7 | Poohla | 8.13 | 4.10 | 0.101 | Calcareous, Slightly saline |
| 8 | Military Cantt. | 7.37 | 4.01 | 0.122 | Calcareous, Slightly saline |
| 9 | Patel Nagar | 7.87 | 2.83 | 0.109 | Calcareous, Salt free |



| 10 | Model Town- 3 | 8.15 | 2.15 | 0.140 | Calcareous, Salt free |
| 11 | Model Town- 1 | 7.95 | 4.73 | 0.092 | Calcareous, Slightly saline |
| 12 | Nachhattar Nagar- a | 7.94 | 3.04 | 0.162 | Calcareous, Salt free |
| 13 | Jaisinghwalla | 8.59 | 4.11 | 0.131 | Alkali or Sodic soils, Slightly saline |
| 14 | Badal | 8.23 | 1.97 | 0.092 | Calcareous, Salt free |
| 15 | Gidderbaha | 8.44 | 4.71 | 0.223 | Alkali or Sodic soils, Slightly saline |
| 16 | Mehmasirja | 8.39 | 6.23 | 0.140 | Alkali or Sodic soils, Slightly saline |
| 17 | Lakhi jungle | 8.45 | 4.12 | 0.109 | Alkali or Sodic soils, Slightly saline |
| 18 | Thermal Colony | 7.87 | 4.28 | 0.070 | Calcareous, Slightly saline |
| 19 | CUPB Campus | 8.31 | 4.70 | 0.092 | Calcareous, Slightly saline |
| 20 | Nachhattar Nagar- b | 7.92 | 4.12 | 0.122 | Calcareous, Slightly saline |
| 21 | Mansa Road near CUPB | 7.94 | 4.02 | 0.131 | Calcareous, Slightly saline |

It was observed that all the 21 soil samples collected from different locations of the study area were mostly alkaline in nature. None of the sampling site was found to be neutral or acidic in nature. The pH range was observed to be lowest in Military Cantt. (pH =7.37) and highest in Jaisingh Walla (pH =8.59) in the study area (Table 5).The value of Electrical Conductivity(EC) ranged between 1.97 dS/m in Badal and 8.78 dS/m in Rampuraphul- a of the study area. 60-70 percent of soil samples were "slightly or moderately saline". Rest of the samples were categorized as "salt free" ones with the values lesser than 4. Based on the pH values, a map for sampling sites as shown in Figure 6(a) was composed in ArcGIS 10.1 software depicting the calcareous and sodic soils at selected sampling sites. On the basis of EC values, salinity of the sampling sites was depicted in the Figure 6(b), which showed that the soils towards the Bathinda city were largely "slightly saline" and few were salt free regions.

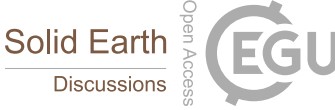

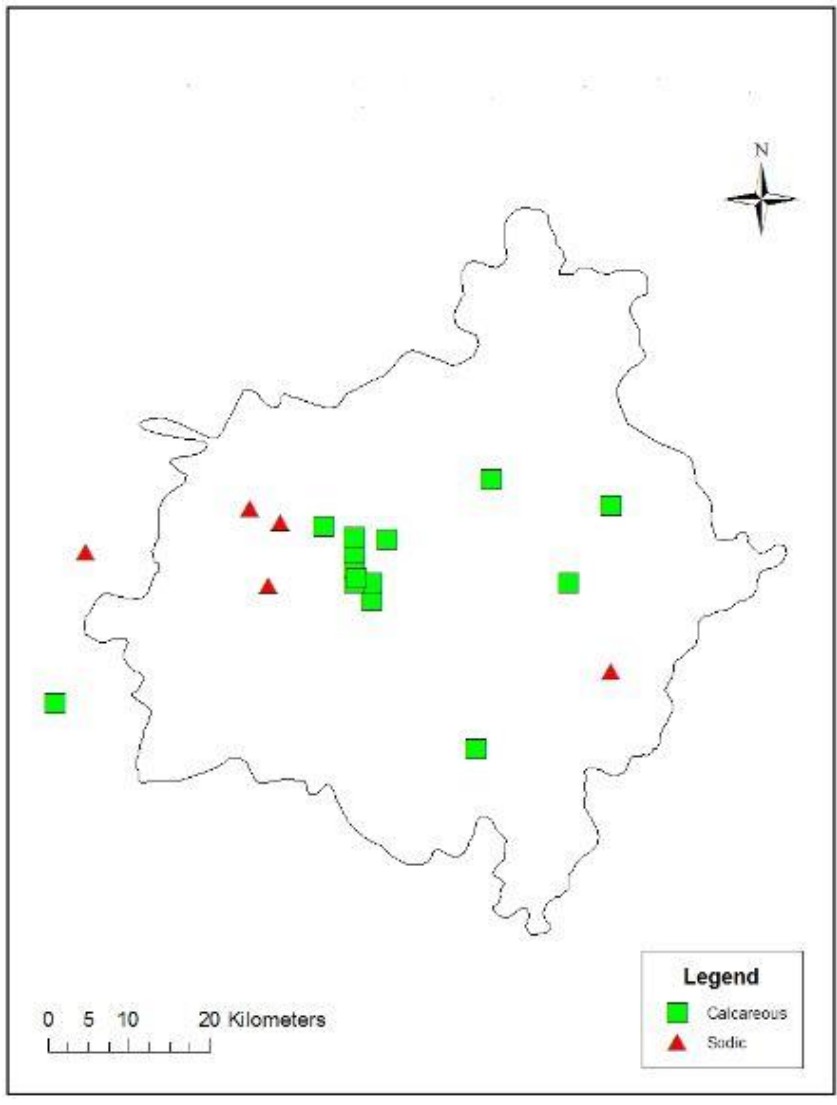

Figure 6(a): Soil types in Bathinda district on the basis of pH values



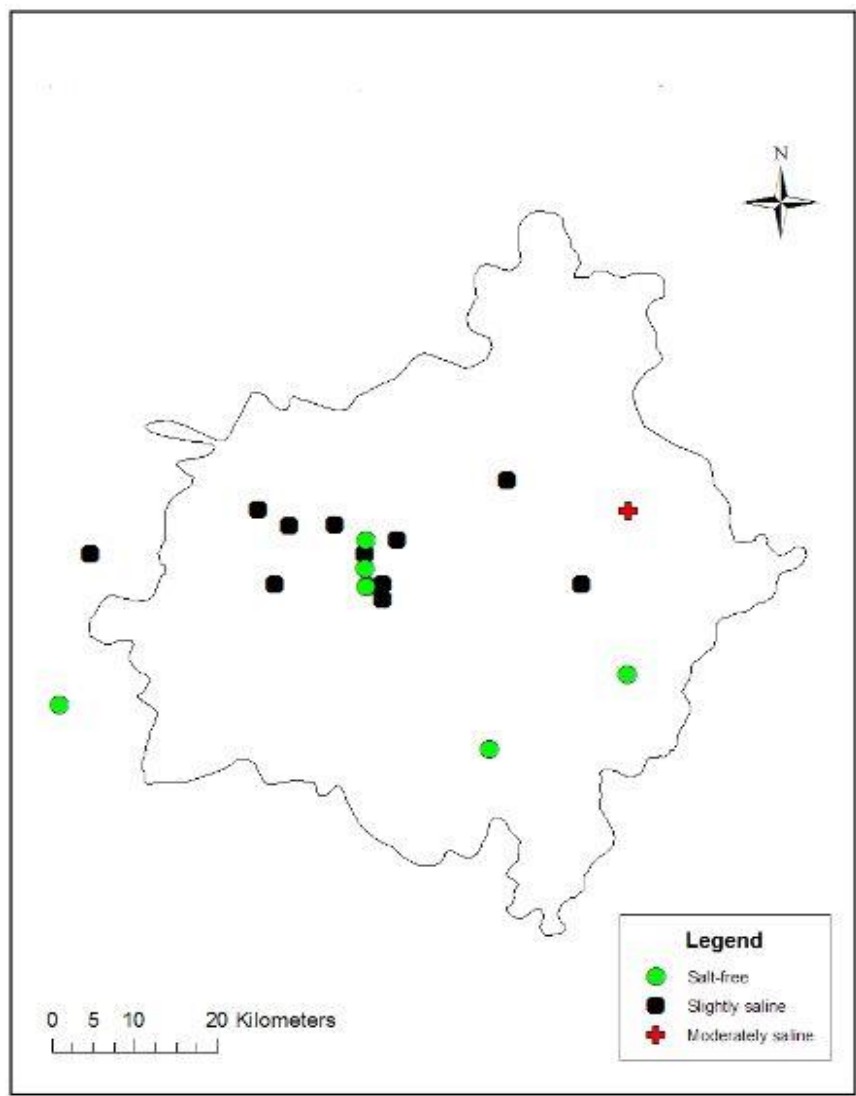

Figure 6(b): Soil types in Bathinda district on the basis of EC values

Phenolphthalein Alkalinity was absent due to the absence of carbonate ions ($CO_3^{2-}$). The alkalinity was then determined by Methyl Alkalinity which indicated the presence of bicarbonate ($HCO_3^-$) ions. The Methyl orange/Total Alkalinity of the soils of the study area was observed to be in the range of 0.070 to 0.223 ($HCO_3^-$) g/L as $CaCO_3$. The results revealed that the saline and alkaline nature of the soils contain larger proportion of sodium, potassium and magnesium and thus, infertile.



## 3.5 Correlation between Physico-Chemical Parameters and Satellite Data Analysis for 2014

Based on the results obtained from laboratory analysis of soil, an attempt was made to establish correlation for band 3 (green), band 5 (NIR) and band 7 (SWIR) of Landsat 8 satellite image for the year 2014 with the pH and Electrical Conductivity of the soil. The Digital Number of corresponding sites was correlated with the physico-chemical parameters.

Table 6: Correlation between band 5 and Physico-Chemical Parameters

|  | DN | pH | EC |
|---|---|---|---|
| DN | 1 | ----- | ----- |
| pH | 0.477 | 1 | ----- |
| EC | 0.285 | -0.003 | 1 |

On the basis of computation of correlation, no significant correlation was observed between physico-chemical parameters and visible (band 3) and short-wave infrared (band 7) of the satellite image; hence, the values have not been reported here. Near infrared band (band 5), however, exhibited significant correlation with physico-chemical parameters compared to rest of the bands; thus, proving it to be a better indicator of soil quality.

## 4. Discussion

A variety of remote sensing data has been used for identifying and monitoring salt-affected areas, including aerial photography, video images, Infrared thermography, visible and infrared multispectral and microwave images (Metternicht and Zinck, 2002).

### 4.1 LULC map of Bathinda: Image Classification and Accuracy Assessment

The digital image classification helped in the identifying, delineating and mapping of the land use/land cover into a number of classes. It has been recommended by various researchers for its potential to detect, map and monitor degradation problems (Sujatha et al., 2000). The classes identified include water bodies, human settlements and built up areas, agricultural land, trees/forest cover and barren land. Multispectral data was used for classification and the categorization on numerical basis related with the spectral pattern of the data for each and every pixel (Lillesand and Kiefer, 1994).

### 4.2 Change detection analysis

The change detection study deals with the comparison of aerial photographs or satellite image of a region taken at different time periods (Petit et al., 2001), performed on a temporal scale to access landscape change caused due to anthropogenic activities on the land (Gibson and Power, 2000). In order to understand landscape patterns for proper land management and decision making improvements; it becomes necessary to consider upon the changes and interactions between human



activities and natural phenomenon (Prakasam, 2010). The remote-sensing change detection has been proven to be a cost-effective method of creating LULC inventories and monitoring land change over time (Coppin et al., 2004; Fry et al., 2011). The results of the present study reveal that area under barren land has decreased from 68.98 km$^2$ in 2000 to 15.26 km$^2$ in the year 2014. This could be due to expansion of settlements and built up areas (Silambarasan et al., 2014) or partly due to

increased vegetation. The settlement expansion on agricultural and forest land is affecting the humans by exposing forest ecosystems and negative impacts of habitat destruction within the forest ecosystem (Stimson et al., 2005). The improper land use is regarded as one of the chief agents of land degradation, which has previously been observed by various researchers (Biro et al., 2013; De Souza et al., 2013; Pallavicini et al., 2014; Mohawesh et al., 2015).However, since settlements and barren land exhibit similar spectral reflectance, slight mixing may have occurred between the two LULC

classes. Zubair (2006) conducted the evaluation of change detection in land use and land cover of Ilorin and its Environs in Kwara State, Nigeria and ultimately concluded that the rapid growth in built-up land was one of the reasons of changing the land use and land cover. The extension of urban land could be a major loss of the cropland (Jin-Song, 2009). While studying the land use pattern in Khed Tehsil of Pune district, Jadhav and Nagarale (2011) also concluded that growth of population, urbanization and transportation network were the socioeconomic factors which changed the land use pattern of Khed Tehsil.

Fazal (2000) in his study on the urban expansion and loss of agricultural land of Saharanpur city, India concluded that rapid conversion of agricultural areas to non-agricultural uses has occurred due to hike in price of land for market operations. Verheye and Paul (1997) revealed that a bulk of households, especially in developing countries, depend on land and other natural resources for satisfying their urgent needs in achieving their long-term aspirations. Such interaction between environment and land use activities show some considerable impacts on the fundamental processes of ecosystem processes

comprising nutrient cycling, soil erosion and human vulnerability (Sala et al., 2000; Foley et al., 2005). Usually the sprawl takes place on urban fringe or along transportation routes (Silambarasan et al., 2014). Such expansion reinforces segregation of income and economic disproportion between urban and suburban communities (Wu, 2006). This high urban sprawl has led to an increased urban land uses, and on the other hand, led to a decreased barren land in the in the form of settlement (Ali, 2011). In the present study, an increase in vegetation was observed in 2014 compared to 2000 which could be due to

recent plantation programmes organized in the region at various educational, institutional and municipal levels. Such plantation may act as 'foster ecosystems' (Lugo, 1988) and accelerates the development of genetic and biochemical diversity on degraded sites (Verma et al., 1982). Sadeghravesh et al. (2016) suggested adopting spatial planning and estimating ecological potential at local, regional and national levels and adapting the applications to land resources to minimize land degradation by way of analyzing desertification strategies using linear assignment method. Further, the strategies to prevent

improper change of land use, development and reclamation of plant cover and controlling overcharging of groundwater resources were identified as the most important strategies for combating desertification.





### 4.3 Laboratory analysis for physico-chemical parameters

The land degradation results in damage to the physical, chemical, or biological properties of the soil which leads to decline in productivity (Chartres, 1987). Thus, it became essential for incorporating the physico-chemical parameters in the present study.

Most of the soil samples were found to be calcareous or saline; while some of the other samples were alkali or sodic in nature (*www.dird-pune.gov.in/phec.htm*).This means that soils contained very high calcium content. The high calcium content keeps the soil in aggregated form and good physical conditions. This is in agreement with the findings of Abbas and Khan (2007) who stated that remotely sensed data integrated with ground truthing verification and physico-chemical analysis is a useful tool for assessment of soil salinity and alkalinity. The remote sensing has been recommended for its potential to
detect, map and monitor degradation problems (Sujatha et al., 2000) including their spread and effect with time (Sommer et al., 1998).

On the basis of EC values given by Ghassemi et al. (1995), it was found that some parts of the study area were slightly saline, some moderately saline and a few areas were salt-free in nature. The saline nature of the soil revealed that the soils contain sodium as soluble salts (usually as $SO_4^{2-}$ or chloride). The soils usually prefer calcium and magnesium over sodium
as a result of which these soils usually have good aeration and aggregate stability (Mathew, 2014). The electrical conductivity and total salt content of a soil extract are the most widely used parameters for describing soil salinity (Liu et al., 2006). Although there are several soil-water ratios used in EC measurements of soil extract, such as 1:1, 1:2, 1:5, 1:10, and also saturation (Zhang et al., 2003), in China 1:5 is the most popular. By the amount of cation exchange capacity (CEC) and percentage of clay, especially high electrical conductivity, the high levels of organic carbon in the low density vegetation
areas (due to destruction of vegetation) are affected (Vagen et al., 2006).

From the results obtained, it could be assessed that the soil is affected by inappropriate activities during past several decades such as loss of soil fertility, erosion, soil salinization etc. Due to the land degradation, vulnerable populations and fragile ecosystems get affected with irreversible results (Bisaro et al., 2014).  The consequences of land degradation may lead to the vegetation loss, soil degradation and pollution of soil, water and air, which need to be addressed to curb the further
degradation (Novara et al., 2013; Batjes et al., 2014; Olang et al., 2014; Srinivasarao et al., 2014). Since the whole state of Punjab is intensively cultivated with 80% of water resources being used for irrigation; the irrigation and overdrafting are some of the main causes of salinization or intrusion of various salts into the soil system (Tiwana, 2007). The ground water irrigation practised predominantly in Bathinda district, is one of the main factors of soil salinization which in turn leads to land degradation (El Baroudy, 2011). The salt remains behind in the soil when water is taken up by plants or lost to
evaporation causing soil salinity (Slinger and Tenison, 2007). The zones receiving low rainfall, shallow water table depth and hot and dry moisture regions in the irrigated areas of the old alluvial plains are found primarily affected by salt soils (Manua and Sharma, 2005). The soils were largely found slightly saline or moderately saline with a few sites found salt free regions. The saline nature of the soil revealed that the soils contain sodium as soluble salts usually as $SO_4^{2-}$ or chloride. The




soils usually prefer calcium and magnesium over sodium as a result of which these soils usually have good aeration and aggregate stability (Mathew, 2014).

Although the degree of salinity was not too high that the soil quality could be affected; but still some remedial measures must be adopted to prevent any problem that might affect the quality of the land as well as the yield in the near future.  In such regions where irrigation is predominantly practised, the remote sensing could be used as a valuable tool for obtaining relevant data on soil salinity (Al Khaier, 2003). Symeonakis et al. (2016) proposed that land degradation being a dynamic process; should not be based on static datasets, rather its assessment should incorporate various temporal datasets while devising parameters for indicators of environmental sensitivity. Thus, according to Behmanesh et al. (2016), various criteria for mapping environmentally sensitive areas were climate, vegetation, soil, groundwater and socio-economic characteristics of land over different time periods.

**4.4 Correlation Analysis**

On the basis of results, the correlation coefficients between soil salinity (Electrical Conductivity) and related Digital Number (DN) values using Landsat data could be helpful in calculating and ascertaining the significant relation between satellite data and soil salinity. The Salt affected soils in arid regions, show a high reflectance, especially when a salt crust (whitish color) is formed. (Alavi-Panah and Goossens, 2001).Further, Mehrjardi et al. (2008) proved that correlation between digital numbers of satellite images and soil salinity could be an efficient parameter for assessing the land degradation by preparing soil salinity maps from remotely sensed data.

**5 Conclusion**

From the results, it was found that the area under human settlements and built up areas expanded by about 10.56%. The land under agricultural practices decreased from 3002.23 km$^2$ to 2399.79 km$^2$. The increased human settlements indicated the alarming growth of human population and the associated increased pressure on land resources due to rising population. The areas under trees/ forest cover increased markedly from 29.4345 km$^2$ to 45.7326 km$^2$. This indicates recent plantation measures adopted by various agencies in the region. The physico-chemical analysis proved to be very useful in assessing the degree of salinization where most of the regions had calcareous soil with slight to moderate salinity and none of the sampling site was found to be highly saline. Correlation studies between the spectral response of the soil and the physico-chemical parameters (pH, EC and alkalinity) revealed that Band 5 was a good indicator of physico-chemical parameters of soil. Thus, it is highly recommended to use multi-temporal, multi-sensor and multi-spectral remotely sensed data for land use / land cover mapping; coupled with GIS to assess the extent of land degradation; which would aid in sound decision making towards restoration of degraded lands and conservation of land resources in an efficient manner. The present study is a useful tool for analysis of environmental sensitivity at regional scale and the identification of hotspots of land degradation. Although the present study does not provide a detailed insight into the causes and manifestations of land degradation;





nevertheless, the present study would help to identify areas vulnerable to land degradation. This would also help in achieving better results with limited investment and avoid wastage of natural resources.

## 6 Acknowledgements

The authors express their gratitude to National Remote Sensing Centre, Hyderabad for providing IRS P6 LISS-III data to carry out the present study. The study was supported by the Central University of Punjab, Bathinda under UGC fellowship for M.Phil.-Ph.D. research scholars (CUPB/MPH-PHD/SEES/EVS/2013-14/17). The authors declare no conflict of interest.

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
