# Peer review of "Assessment and Monitoring of Land Degradation Using Geospatial Technology in Bathinda District, Punjab, India"

_Solid Earth, 2016_

## Referee Comment (RC1) · F. Pacheco (Referee) · 15 Jan 2017

REVISION Paper: se-2016-172 Title: Assessment and Monitoring of Land Degradation Using Geospatial Technology in Bathinda District, Punjab, India

OUTLINE AND GENERAL APPRECIATION This study correlates remote sensing data (digital numbers) with physical-chemical parameters of soils (e.g. salinity) in the Bathinda District, Punjab, India, with the main purpose of using the correlations to assess land degradation across larger areas through soil salinity maps prepared from remotely sensed data. The study is interesting but needs some improvements (mostly detailed descriptions on several sections) before being prepared for publication, as detailed below.

CONCERNS 1) The study objectives are not clearly stated. The authors should also mention the novelty brought by this study, when compared to other similar studies. 2) In the introduction, the authors state that "Improper land use practice has been attributed as one of the major causes of land degradation by various researchers". A major form of improper land use is one leading to environmental land use conflicts. This type of improper use is characterized by a deviation between the actual and natural uses set by land capability. Environmental land use conflicts and their consequences for land degradation have been recently studied by various authors, namely (Pacheco et al., 2014, 2016; Valera et al., 2016; Valle Junior et al., 2014a,b; 2015). A mention to these studies would be appropriate. 3) The description of the study is is rather short. What are the reasons of soil salinizationin the studied area? Is that irrigation? Of what cultures? Since when? What are the main crops and management practices in the area? No basic information is provided in the study area description, which should appear in the revised version. 4) Did the authors tried to use images free from meteorological effects, like MODIS? 5) What type of correlation analysis was used? Was it based on time series of DN or on average values? How was seasonality incorporated in the correlation analysis? A more detailed description of the results is required. 6) As with the study area description, the presentation of methods is also fragile. The authors must improve this section in the revised version.

RECOMMENDATION Major revision 14 January 2017

REFERENCES Pacheco, F.A.L., Varandas, S.G.P., Sanches Fernandes, L.F., & Valle Junior, R.F. (2014). Soil losses in rural watersheds with environmental land use conflicts. Science of the Total Environment, v. 485–486C, p. 110–120. Valle Junior, R.F., Varandas, S.G.P., Sanches Fernandes, L.F., & Pacheco, F.A.L. (2014). Environmental land use conflicts: A threat to soil conservation. Land Use Policy, v. 41, p. 172–185. Valle Junior, R.F., Varandas, S.G.P., Sanches Fernandes, L.F., & Pacheco, F.A.L. (2014). Groundwater quality in rural watersheds with environmental land use conflicts. Science of the Total Environment, v. 493, p. 812–827. Valle Junior, R.F., Varandas,

S.G.P., Pacheco, F.A.L., Pereira, V.R., Santos, C.F., Cortes, R.M.V., Fernandes, L.F.S. (2015). Impacts of land use conflicts on riverine ecosystems. Land Use Policy, v. 43, p. 48-62. Pacheco F.A.L., Sanches Fernandes, L.F. (2016). Environmental land use conflicts in catchments: a major cause of amplified nitrate in river water. Science of the Total Environment, v. 548–549, p. 173–188. Valera, C.A., Valle Junior, R.F., Varandas, S.G.P., Sanches Fernandes, L.F., Pacheco, F.A.L. (2016). The role of environmental land use conflicts in soil fertility: A study on the Uberaba River basin, Brazil. Science of the Total Environment, v. 562, p. 463–473.
* * *

---

## Referee Comment (RC2) · SA ADENIYI (Referee) · 21 Feb 2017

Review of research paper: se-2016-172 " Assessment and Monitoring of Land Degradation Using Geospatial Technology in Bathinda District, Punjab, India"

This study aim to integrate remote sensing data and field-based soil data to assess severity of land degradation in the Bathinda District, Punjab. The authors selected multispectral Landsat 7 and 8 images for 2000 and 2014 to conduct a land use and land cover change of the study area. Next, soil data analysed for three physico-chemical parameters of soil quality (i.e pH, EC and Alkalinity) collected from 21 sites within the study area were correlated with remotely sensed data, particularly band 5 of the pixels. For me, the images were well processed and the result of Land use/land cover

change is commendable. Without any iota of doubt, the study represents a great effort on the part of the authors to use geospatial technology in natural resource management. Great work. However, the paper needs some improvements before being recommended for publication. I therefore, listed specific areas that require authors attention as follows:

Title: I considered the title ok. Abstract: The abstract is well written. However page 1, line 8 , how was severity measured in this study. This should be defined in quantitative term and should be mentioned in the abstract. The abstract is a synopsis of the whole research. Introduction The introduction needs improvement. Start with line 27 and move lines 23-26 to other part of the introduction. The title suggests that we are dealing with land degradation and not land use/land cover change per se. This should reflect in the introduction. What is the gap in knowledge and how does this paper fill that gap? Is the contribution of this paper to knowledge in term of methodology or what? Having gone through the whole paper, I guess the major issue is in the area of methodology. (i.e Integration of remotely sensed data with field based data to determine the severity of land degradation). This can be included in the introduction Study Area Detailed description of the research context is required at this stage, particularly to help readers who are not familiar with the study area. Figure 1 as indicated in the paper now is on data collection issues and not the study area per se. The location map needs to show the location of Bathinda District, Punjab, in relation to India. The main map should be Bathinda District, and the smaller map should be India. The location map is meant to orientate the reader. Methodology Data collection and analysis What informed the choice of Land Sat 7 and 8 images used and the year selected? Please discuss? I see that the soil samples were analysed for chemical properties only, and not physico-chemical soil properties. Why only three soil quality parameters? One would have expected that more soil quality parameters be included in the analysis. Here, I suggest that textural properties as well. Next, why 21 soil sample points? Was this a function of cost or time? Please argue this out . Also give the coordinates of the sample points in table 5. Data Analysis To me, there is a confusion on the table of correlation. It is important to clarify whether DN used in the correlation matrix is the same as values of band 5 (Near infrared band) of the pixels for 2014 image. The correlation analysis needs to be tested statistically at 0.5 level of significance. The significance of the correlation coefficients between DN/pH; EC/pH,DN/EC should be tested for significance here. Result and discussion Page 21 line 15-17 "Mehrjardi et al. (2008) proved that correlation between digital numbers of satellite images and soil salinity could be an efficient parameter for assessing the land degradation by preparing soil salinity maps from remotely sensed data" . A fundamental issue that I think is lacking in this paper, and which I believe readers of this work would like to see is a map shown the severity of land degradation. This is missing and can be done within the GIS environment using geostatistical analyst in Arc GIS. This appears to be the missing link in the paper and should be included in the revised version of the paper. There are several methods to do this. The authors may want to use any of the known interpolation methods. Kriging may be a good method to estimate the variable (salinity) over space. Then, check the validity of the model using the analysis of variance with the related error means and the mean square of error. ARCMap can be effectively used to analyze the spatial structure of the data and to define the semivariograms. See the following: 1. Burgess, T. M., and R. Webster. 1980. Optimal interpolation and isarithmic mapping of soil properties: The variogram and punctual kriging. Journal of Soil Science 31:315–331 2. Alejandra Mora-Vallejo et. Al 2008 Small scale digital soil mapping in Southeastern Kenya . Catena 76: 44-53 3. Behera, S.K and Shukla, A K ( 2014) Spatial Distribution of Surface Soil Acidity, Electrical Conductivity, Soil Organic Carbon Content and Exchangeable Potassium, Calcium and Magnesium in Some Cropped Acid Soils of India Land degradation and Development . DOI: 10.1002/ldr.2306

4. Sheng et al (2010) Digital soil mapping to enable classification of the salt-affected soils in desert agro-ecological zones Agricultural Water Management 97: 1944–1951

General Impression: Presently, the paper focuses more on the analysis of land use and land cover change (LULC) in Bathinda District, Punjab, India. This is the strength of the

paper for now and efforts of the authors commendable. However, the issue of severity of land degradation has been glossed over. Severity of land degradation should be shown in quantitative terms within the GIS environment. I hope I have provided suggestions that could be used to improve the quality of this paper. Based on the above observations, I wish congratulate the authors for good job and also recommend a major revision on this paper before publication.

---

## Referee Comment (RC3) · Anonymous Referee #3 · 7 Mar 2017

Dear Editor of SE

I'm sending my comments to the Editor regarding the manuscript se-2016-172 "Assessment and Monitoring of Land Degradation Using GeospatialTechnology in Bathinda District, Punjab, India" by Naseer Ahmad and Puneeta Pandey

First of all I would like to congratulate the author for the initiative of mapping the evolution of the Bathinda District land use all over the last 14 years. In respect to the review, I would like to highlight the following aspects:

1. Does the paper address relevant scientific questions within the scope of SE? Yes, but not as presented.

[Figure]

2. Does the paper present novel concepts approaches, ideas, tools, or data? The novelty is not clear in the text. 3. Are substantial conclusions reached? See below. 4. Are the scientific methods and assumptions valid and clearly outlined? Yes. 5. Are the results sufficient to support the interpretations and conclusions? See below. 6. Is the description of experiments and calculations sufficiently complete and precise to allow their reproduction by other scientists (traceability of results)? See below. 7. Do the authors give proper credit to related work and clearly indicate their own new/original contribution? See below. 8. Does the title clearly reflect the contents of the paper? The manuscript shows the modifications in the land use and land cover within a 14 years interval but doesn't present enough the land degradation attributes and statements. 9. Does the abstract provide a concise and complete summary? See below. 10. Is the overall presentation well structured and clear? See below. 11. Is the language fluent and precise? Yes it is. 12. Are mathematical formulae, symbols, abbreviations, and units correctly defined and used? 13. Should any parts of the paper (text, formulae, figures, tables) be clarified, reduced, combined, or eliminated? 14. Are the number and quality of references appropriate? The authors should improve the quality of references including basic-scientific papers.

General comments:

- Although the effort of the authors was great in using geospatial technology for mapping the natural resources over the years, while evaluating the paper it was not clear the monitoring of land degradation indicated in the title and along the text. The main comments points are listed below: - The study just shows the land cover and land use evolution within 14 years but doesn't connect that information with land degradation. Is there any correlation between land degradation and land cover? Also we can't see correlation between the soil properties and land degradation. The scientific question can be: how was land degradation studied yearly and how may this Âĺnew approachÂĺ improve it? - As said in the abstract, the aim of the study was to assess land degradation with the help of geospatial technology Remote Sensing (RS) and Geographical

Information System (GIS) in Bathinda District, Punjab. All over the paper we can't see the extension and the reasons of land degradation for the evaluated period. What is the novelty of the present study? - The conclusion that the soils in the study area were exposed to the salt intrusion is based on the 21 samples; hence, the authors should better explain the sampling methodology because the samples are not well distributed all over the mapped region. - Another weak point in the MS is about the severity of land degradation: how can you measures the severity parameters? - The bibliographic review must be improved by mentioning studies that used the same techniques – RS, GIS and physic-chemical analysis of soil – to evaluate land degradation and the reasons why those techniques were chosen.

Finally, I cannot recommend this manuscript for publication in its present form. A major revision is needed to improve the manuscript.

Best regards,

Reviewer

---

## Author Comment (AC1) · 18 Apr 2017

Answers to Comments of RC1 1) Comments of Referee: The study objectives are not clearly stated. The authors should also mention the novelty brought by this study, when compared to other similar studies. Author's Response: Objectives: This study aim to integrate remote sensing data and field-based soil data to assess severity of land degradation in the Bathinda District, Punjab. Novelty: Integration of remotely sensed data with field based data to determine the severity of land degradation is an important aspect of the present study. However, as far as land degradation assessment through geospatial technology is concerned, not much study has been done in the Bathinda region of Punjab, a north-western state of India. Author's

changes in the manuscript: Suggestions incorporated in line no. 2-4 of Page 4. 2) Comments of Referee: In the introduction, the authors state that "Improper land use practice has been attributed as one of the major causes of land degradation by various researchers". A mention to the studies of Pacheco et al., 2014, 2016; Valera et al., 2016; Valle Junior et al., 2014a,b; 2015would be appropriate. Author's Response: Improper land use practice has been attributed as one of the major causes of land degradation by various researchers (Biro et al., 2013; De Souza et al., 2013; Pallavicini et al., 2014; Mohawesh et al., 2015). A major form of improper land use is the one leading to environmental land use conflicts that develop on soils used for activities not in compliance with the natural potential of the soil. This type of improper use is characterized by a deviation between the actual and natural uses set by land capability. Environmental land use conflicts and their consequences for land degradation have been recently studied by various authors, namely (Pacheco et al., 2014, 2016; Valera et al., 2016; Valle Junior et al., 2014a,b; 2015). Author's changes in the manuscript: Suggestions incorporated in line no. 12-17 of Page 2. 3) Comments of Referee: The description of the study is is rather short. What are the reasons of soil salinization in the studied area? Is that irrigation? Of what cultures? Since when? What are the main crops and management practices in the area? No basic information is provided in the study area description, which should appear in the revised version. Author's Response: The study area was further described as per the comment received. So far as salinization is concerned a few studies (Sheng, 2010; El Baroudy, 2011; Koshal, 2012) were mentioned regarding the reasons of salinization. The crops grown in the study area and the management practices were also incorporated. Author's changes in the manuscript: Line 16-29 of Page 4 incorporates the above mentioned suggestions. 4) Comments of Referee: Did the authors tried to use images free from meteorological effects, like MODIS? Author's Response: No, MODIS images were not used for the present study. However, Landsat images used in the present study were cloud-free scenes. Author's changes in the manuscript: Not required 5) Comments of Referee: What type of correlation analysis was used? Was it based on time series

of DN or on average values? How was seasonality incorporated in the correlation analysis? A more detailed description of the results is required. Author's Response: Pearson's correlation analysis was used in the present study. It was not based on time series of DN or on average values; rather the DN value corresponding to geographical coordinates of sampling sites was considered for the present study. Author's changes in the manuscript: The table 6 describes the correlation analysis. Some text has been revised and presented in line 6-8 of Page 21. 6) Comments of Referee: As with the study area description, the presentation of methods is also fragile. The authors must improve this section in the revised version. Author's Response: The revised methodology and flowchart of methodology has been added in Section 2.6 of Materials and Method. Author's changes in the manuscript: Section 2.6 of lines 14-22 of Page 7.

Please also note the supplement to this comment:
http://www.solid-earth-discuss.net/se-2016-172/se-2016-172-AC1-supplement.pdf

———————————————————

---

## Author Comment (AC2) · 18 Apr 2017

Answers to Comments of Referee 2 1. Comments of Referee: Abstract: The abstract is well written. However page 1, line 8 , how was severity measured in this study. This should be defined in quantitative term and should be mentioned in the abstract. The abstract is a synopsis of the whole research. Author's Response: The severity of land degradation was estimated quantitatively by analysing the physico-chemical parameters in the laboratory to determine saline/salt free and calcareous/sodic and further correlating them with satellite based studies. The pH varied between 7.37 to 8.59; Electrical Conductivity (EC) between 1.97 dS/m and 8.78 dS/m and the Methyl orange/Total Alkalinity area in the range of 0.070 to 0.223 (HCO3-) g/L as CaCO3. The

spatial variability of these soil parameters have been depicted through soil maps generated in GIS environment. Author's changes in the manuscript: Incorporated in lines 8-13. 2. Comments of Referee: Introduction The introduction needs improvement. Start with line 27 and move lines 23-26 to other part of the introduction. Author's Response: The introduction has been improved according to the Referee's comments in the revised version. The Introduction section starts with line 27 as suggested. Author's changes in the manuscript: Line 27 has been moved to the beginning of Introduction as Line 26-29 followed by rest of the Introduction. 3. Comments of Referee: The title suggests that we are dealing with land degradation and not land use/land cover change per se. This should reflect in the introduction. Author's Response: Line 29 of Page 1 to Lines 1-6 of Page 2 talk about land degradation and land use change. Author's changes in the manuscript: No corrections made. 4. Comments of Referee: What is the gap in knowledge and how does this paper fill that gap? Author's Response: Integration of remotely sensed data with field based data to determine the severity of land degradation is an important aspect of the present study. However, as far as land degradation assessment through geospatial technology is concerned, not much study has been done in the Bathinda region of Punjab, a north-western state of India. With the onset of GIS technology, mitigation of soil degradation could be handled by land suitability spatial models and land use change. A link can be established between land use change and land degradation; as well as land use change and spatial distribution of soil contaminants through geostatistical and non-geostatistical interpolation methods in GIS environment. Therefore, an attempt was made to fulfill the gap by using interpolation method for assessment of land degradation severity and the correlation between digital number (DN) of satellite data and the parameters of the soil. The present study was carried out with the objective of integrating remote sensing data and field-based soil data to assess severity of land degradation in the Bathinda District, Punjab. Author's changes in the manuscript: Section 1.1 of Page 4 (Lines 1-11).

5. Comments of Referee: Is the contribution of this paper to knowledge in term of methodology or what? Having gone through the whole paper, I guess the major issue is in the area of methodology. (i.e Integration of remotely sensed data with iňẠeld based data to determine the severity of land degradation). This can be included in the introduction. Author's Response: Section 2.6 and 2.7 in methodology describes the integration of remotely sensed data with field based data to determine the severity of land degradation. A brief paragraph about the integration of remotely sensed data with the field based data such as pH EC and alkalinity was discussed in Section 2.6 regarding the determination of severity of land degradation using Inverse Distance Weighted (IDW) interpolation method. Section 2.7 Author's changes in the manuscript: Line 12-21 of Page 6 (Section 2.6) 6. Comments of Referee: Study Area Detailed description of the research context is required at this stage, particularly to help readers who are not familiar with the study area. Author's Response: A detailed description of the study area with the environmental problems persisting in the area have been added in Section 2.1 (Lines 22-31, page-3). Author's changes in the manuscript: The text has been included in lines 22-31 of Page 3. 7. Comments of Referee: Figure 1 as indicated in the paper now is on data collection issues and not the study area per se. The location map needs to show the location of Bathinda District, Punjab, in relation to India. The main map should be Bathinda District, and the smaller map should be India. The location map is meant to orientate the reader. Author's Response: The study area map was corrected accordingly with a larger map of Bathinda and a smaller one of Punjab and then India. Author's changes in the manuscript: The location map has been revised as suggested and presented as Figure 1. 8. Comments of Referee: Methodology Data collection and analysis What informed the choice of Land Sat 7 and 8 images used and the year selected? Please discuss? Author's Response: The satellite image Landsat 8 was chosen for the study for two reasons: to prepare land use land cover (LULC) map at reasonably good spatial scale (resolution of Landsat 8 in visible and near infrared (NIR) band is 30 m and 60m respectively); and also for correlation analysis, it was necessary that the time lag between field sampling and satellite data procurement should be as minimum as possible. Landsat 7 was chosen for change detection studies because decadal change detection required older dataset to get most
up-to-date change detection output. Author's changes in the manuscript: No change done. 9. Comments of Referee: I see that the soil samples were analysed for chemical properties only, and not physico-chemical soil properties. Why only three soil quality parameters? One would have expected that more soil quality parameters be included in the analysis. Here, I suggest that textural properties as well. Author's Response: The three soil parameters were selected based on the problems of soil persisting in the study area. However, I do agree that textural properties could have been included, but it is not possible at this stage. We shall take this suggestion into consideration for future research. Author's changes in the manuscript: None 10. Comments of Referee: Next, why 21 soil sample points? Was this a function of cost or time? Please argue this out. Author's Response: 21 sample points were selected based on land use land cover features present in the study area. Besides, the study was a short term one (carried out over a period of six months); hence, it could be considered a function of time. Author's changes in the manuscript: None

11. Comments of Referee: Also give the coordinates of the sample points in table 5. Author's Response: The coordinates of sampling points have been added to Table 5. Author's changes in the manuscript: Table 5 of Page 16-17 incorporates the geographical coordinates as well.

12. Comments of Referee: Data Analysis To me, there is a confusion on the table of correlation. It is important to clarify whether DN used in the correlation matrix is the same as values of band 5 (Near infrared band) of the pixels for 2014 image. The correlation analysis needs to be tested statistically at 0.5 level of significance. Author's Response: The DN values used in the correlation matrix are the same as the values of near infrared band (Band 5) for the geographical coordinates of the sampling sites. The correlation analysis carried out was Pearson's correlation analysis at 0.5 level of significance. 13. Comments of Referee: The significance of the correlation coefficients between DN/pH; EC/pH,DN/EC should be tested for significance here. Author's Response: The correlation analysis between band 5 and pH and EC values

has been given in Table 6. Author's changes in the manuscript: The correlation analysis has been already given as Table 6. Discussion for the same has been given in lines 11-17 of Page 22.

14. Comments of Referee: A fundamental issue that I think is lacking in this paper, and which I believe readers of this work would like to see is a map shown the severity of land degradation. This is missing and can be done within the GIS environment using geostatistical analyst in Arc GIS. This appears to be the missing link in the paper and should be included in the revised version of the paper. Author's Response: Section 3.5 of the manuscript deals with soil map showing severity of land degradation in GIS environment. Digital soil mapping was used for the prediction of spatial variability of individual soil properties in large areas over space, where maps are generated in digital format in a rapid, effective, efficient, and low cost manner (Sheng, 2010). Severity of land degradation was shown as spatial distribution of pH, EC and Alkalinity in quantitative terms via Inverse Distance Weighted (IDW) interpolation methods using statistical analyst in Arc GIS. Based on the pH values, a soil map for sampling sites as shown in Figure 6(a) was composed in ArcGIS 10.1 software. Similarly on the basis of EC and Alkalinity values, soil maps were composed respectively as shown in Figure 6(b) and 6 (c) depicting the severity of land degradation in terms of salinity. Author's changes in the manuscript: Suggestions incorporated in Section 3.5 (Page 17-19).

---

## Author Comment (AC3) · 18 Apr 2017

Answers to Comments of Referee 3

1. Comments of Referee: Are the number and quality of references appropriate? The authors should improve the quality of references including basic-scientific papers. The bibliographic review must be improved by mentioning studies that used the same techniques – RS, GIS and physic-chemical analysis of soil – to evaluate land degradation and the reasons why those techniques were chosen. Author's Response: The new references have been added at relevant places in Introduction section. Author's changes in the manuscript: Lines 3-15 and 19-27 of Page 3 describe the references.

[Figure]

2. Comments of Referee: The study just shows the land cover and land use evolution within 14 years but doesn't connect that information with land degradation. Is there any correlation between land degradation and land cover? Also we can't see correlation between the soil properties and land degradation. Author's Response: Yes indeed there is a correlation between land degradation and land cover. The link between the two has been incorporated in the revised version between line no 3-16 of Page 3 in the introduction. Author's changes in the manuscript: Line no 3-16 of Page 3 in the Introduction section. 3. Comments of Referee: The scientific question can be: how was land degradation studied yearly and how may this new approach improve it? - As said in the abstract, the aim of the study was to assess land degradation with the help of geospatial technology Remote Sensing (RS) and Geographical Information System (GIS) in Bathinda District, Punjab. All over the paper we can't see the extension and the reasons of land degradation for the evaluated period. Author's Response: The land degradation was not studied yearly, rather it was studied for the year 2014 only based on satellite data and laboratory analysis. For decadal change detection, satellite data for the year 2000 was used to understand the changes the land might have undergone through. Author's changes in the manuscript: Page 29 (Line 13-31) and Page 30 (line 1-2) discuss these attributes of the study area. 4. Comments of Referee: What is the novelty of the present study? - The conclusion that the soils in the study area were exposed to the salt intrusion is based on the 21 samples; hence, the authors should better explain the sampling methodology because the samples are not well distributed all over the mapped region. Author's Response: The soil samples were selected randomly and equally distributed on the study area. As regards novelty, integration of remotely sensed data with field based data to determine the severity of land degradation is an important aspect of the present study. However, as far as land degradation assessment through geospatial technology is concerned, not much study has been done in the Bathinda region of Punjab, a north-western state of India. Author's changes in the manuscript: Suggestions incorporated in line no. 2-4 of Page 4.